# Gut-Modulating Agents and Amyotrophic Lateral Sclerosis: Current Evidence and Future Perspectives

**DOI:** 10.3390/nu16050590

**Published:** 2024-02-21

**Authors:** Ahmed Noor Eddin, Mohammed Alfuwais, Reena Noor Eddin, Khaled Alkattan, Ahmed Yaqinuddin

**Affiliations:** College of Medicine, Alfaisal University, Riyadh 11533, Saudi Arabia; mohfuwaiz@gmail.com (M.A.); neddin.reena@gmail.com (R.N.E.); kkattan@alfaisal.edu (K.A.); ayaqinuddin@alfaisal.edu (A.Y.)

**Keywords:** amyotrophic lateral sclerosis, neurodegeneration, microbiota, gut-brain axis, gut modulation, dysbiosis, probiotics, therapeutics

## Abstract

Amyotrophic Lateral Sclerosis (ALS) is a highly fatal neurodegenerative disorder characterized by the progressive wasting and paralysis of voluntary muscle. Despite extensive research, the etiology of ALS remains elusive, and effective treatment options are limited. However, recent evidence implicates gut dysbiosis and gut–brain axis (GBA) dysfunction in ALS pathogenesis. Alterations to the composition and diversity of microbial communities within the gut flora have been consistently observed in ALS patients. These changes are often correlated with disease progression and patient outcome, suggesting that GBA modulation may have therapeutic potential. Indeed, targeting the gut microbiota has been shown to be neuroprotective in several animal models, alleviating motor symptoms and mitigating disease progression. However, the translation of these findings to human patients is challenging due to the complexity of ALS pathology and the varying diversity of gut microbiota. This review comprehensively summarizes the current literature on ALS-related gut dysbiosis, focusing on the implications of GBA dysfunction. It delineates three main mechanisms by which dysbiosis contributes to ALS pathology: compromised intestinal barrier integrity, metabolic dysfunction, and immune dysregulation. It also examines preclinical evidence on the therapeutic potential of gut-microbiota-modulating agents (categorized as prebiotics, probiotics, and postbiotics) in ALS.

## 1. Introduction

Amyotrophic lateral sclerosis (ALS), also known as Lou Gehrig’s disease, is a rare but highly fatal neurodegenerative disorder characterized by the progressive loss and degeneration of motor neurons in the brain and spinal cord. Its average age of onset is currently estimated at 61.8 years, with death typically occurring 3–5 years following diagnosis [1,2]. Although the incidence of ALS is around 2.0 cases per 100,000 persons per year, its global prevalence is closer to 5.4 cases per 100,000 persons, reflecting a low survival rate [2]. The clinical manifestations of ALS are heterogeneous and can be initially non-specific. Patients afflicted with the disease often experience debilitating muscle weakness, stiffness, or paralysis, which begin in the extremities and then spread to involve most parts of the body. Despite extensive research efforts, the etiology of ALS remains largely unknown, and thus effective treatment options that can delay disease progression are limited. However, a growing body of evidence has recently implicated enteric microbiota and the gut–brain axis (GBA), an intricate two-way communication system between the gastrointestinal (GI) tract and central nervous system (CNS), in the pathogenesis of several neurodegenerative diseases, including ALS [3].

Factors contributing to overall gut health, such as intestinal inflammation and barrier permeability, can have profound effects on brain function and well-being [3,4]. Symptoms of GI upset or dysfunction (e.g., pain, dysphagia, reflux, and constipation) have also been noted in ALS, further suggesting a link between gut and brain pathology [5,6,7]. Moreover, imbalances or alterations in the composition of the gut microbiome, commonly referred to as gut dysbiosis, have been observed in human patients as well as in experimental animal models of ALS [8,9]. Although several advancements in our understanding of the gut microbiome and its relation to neurodegenerative disorders have been made, the field is still in its early stages. Further explorations into the complex role of the GBA and its contributions to brain development in both health and disease can provide new insights into the mechanisms underlying ALS pathogenesis. In this review, we comprehensively summarize and present the current literature on gut dysbiosis in ALS, with a specific focus on the involvement of GBA pathology. Moreover, we aim to explore the potential of direct gut modulation as a therapeutic measure in the management of this devastating motor disease.

## 2. Components of the Gut–Brain Axis

Bidirectional communication between the gut and brain is primarily mediated by the nervous, immune, and endocrine pathways [10,11]. Nervous pathways include the enteric nervous system (ENS), autonomic nervous system (ANS), and vagus nerve. Visceral afferents normally transmit sensory information from the gut to the CNS, which, in turn, modulates GI function by efferent signals. Intestinal motility, absorption, secretion, and blood flow are all regulated by the ANS [11]. Interestingly, the ANS shares several mediators and receptors with the gut’s immune system and, thus, can also play a role in regulating local inflammatory processes [12]. Signaling in the immune pathways is primarily mediated by enterocytes, resident immune cells, and the gut microbiota. Enterocytes, particularly a specialized subset called tuft cells, and gut-associated immune cells, such as macrophages, neutrophils, and dendritic cells, all express toll-like receptors (TLRs) that can recognize molecular patterns on the surfaces of invading pathogens and initiate an innate immune response [11]. Upon activation, these cells release inflammatory cytokines and chemokines that not only aid in further immune cell recruitment but also trigger signaling cascades that can communicate with the CNS. Cytokines may also act locally on vagal afferents [10] and subsequently affect signaling within the GBA. Endocrine pathways mainly involve hormones and the hypothalamic–pituitary–adrenal (HPA) axis. However, neuroactive microbial products, such as short-chain fatty acids (SCFAs) and secondary bile acids (2-BAs), can also propagate signals to the CNS either directly, by accessing the body’s systemic circulation, or indirectly, by interacting with enteroendocrine, enterochromaffin, and immune cells of the gut [10]. While this section offers a simplified overview of these pathways, it is important to recognize how different components of the GBA are related and work in concert to maintain gut homeostasis. Furthermore, it becomes evident that disruption in any component of the GBA can not only impact the gut but may also have downstream effects on the CNS.

## 3. Evidence of Gut Dysbiosis in Human Amyotrophic Lateral Sclerosis

The human gut is habitat to a highly complex yet balanced network of commensal microorganisms spanning all three domains of life: bacteria, archaea, and eukarya (including fungi, yeasts, and protozoa) [13]. Bacteriophages and eukaryotic viruses are also integral components of the gut microbiome [14]. Although the exact composition and diversity of microbial communities throughout the GI tract vary considerably, the gut microbiome is primarily dominated by bacterial populations. Specifically, among the eight identified bacterial phyla in the human gut, Firmicutes and Bacteroidetes constitute the majority portion and represent over 90% of all intestinal microbiota [13,15]. The remaining 10% is typically composed of Actinobacteria and smaller proportions of Proteobacteria, Verrucomicrobia, or Cyanobacteria [13,15,16]. The division of the intestinal flora at this taxonomic level is similar, if not uniform, across most healthy humans. However, each individual possesses a unique microbiome made up of different species and strains at varying densities likely owing to a number of genetic factors and host–microbe interactions [17,18]. Age, diet, lifestyle, environment, and disease may also account for individualized differences since they can shape and alter the gut’s microbial composition [15]. Furthermore, studies have shown that dietary changes or acute lifestyle modifications can quickly and reproducibly alter the gut microbiota [19,20,21]. Over the past decade, the relationship between gut health and neurological disorders has become a subject of research interest. Preclinical evidence implicates gut dysbiosis and subsequent GI dysfunction in the pathogenesis of ALS [22,23,24,25]. In parallel, human studies consistently report that the abundance and diversity of the intestinal flora in ALS patients are significantly altered when compared to healthy or other neurodegenerative controls. 

Alterations to the gut flora are typically examined via high-throughput sequencing techniques such as 16S rRNA sequencing and shotgun metagenomics. Bioinformatic analyses may also be used to assess the taxonomic composition or diversity of detected microbial communities in various samples, as well as across different patient groups. To date, the largest human gut profiling study was conducted by Guo et al. [26] and included a total of 185 participants. The fecal microbiome of ALS patients and unrelated healthy controls were longitudinally compared at three different time points. Two bacterial phyla, Firmicutes and Cyanobacteria, were significantly different in relative abundance between patients and controls at the first collection point (baseline) [26]. Adjustment for confounding factors (sex, age, and body mass index) further highlighted alterations to the abundance of six specific genera: *Bacteroides*, *Parasutterella*, and *Lactococcus* were all significantly enriched in ALS samples, while *Faecalibacterium* and *Bifidobacterium* were markedly reduced compared to controls. The presence of a distinct gut microbiome in ALS patients was further validated by significant differences in the beta-diversity between the two study groups. Interestingly, only the abundance of Firmicutes significantly varied at the second time point [26], suggesting that the gut microbiome may continue to undergo changes over the course of the disease. In tandem, the relative abundance of Bacteroidetes and Firmicutes was significantly different at the first and second time points, respectively, in ALS patients with bulbar versus limb onset. Separate cohorts, albeit with some overlap in study participants, were also used to assess if any associations between dysbiosis and plasma metabolites were present [26,27]. Indeed, several microbes, such as *Akkermansia muciniphila* and members of the *Lachnospiraceae* family were correlated with alterations in plasma lipid-related metabolites [26]. Mendelian randomization analysis described acylcarnitine, bile acid, and fatty acid metabolism as potentially causal to ALS, and several acylcarnitines were further negatively correlated with scores on the revised ALS functional rating scale (ALSFRS-R). Taken all together, the microbiome-metabolome interface offers a promising framework for understanding the disease mechanisms underlying ALS and exploring new therapeutic interventions. 

Other human microbiome studies also provide evidence of dysbiosis in ALS patients. For instance, Fang et al. [28] demonstrated several significant alterations to the gut microbiome at different taxonomic levels. Bacteroidetes (phylum), Bacteroidia (class), Bacteroidales (order), and *Dorea* (genus) were all enriched in ALS samples compared to healthy controls, whereas Firmicutes (phylum), Clostridia (class), *Lachnospiraceae* (family), *Oscillibacter* (genus), and *Anaerostipes* (genus) were notably decreased. Di Gioia et al. [29] reported that *Escherichia coli*, *Clostridiales Family XI* (family), *Gastranaerophilalaes* (family), and Cyanobacteria (phylum) were all significantly elevated in ALS, while *Clostridiaceae 1* (family) was lower in patients than controls. While the total bacterial count did not differ between the two study groups, ALS stool samples showed lower DNA concentrations compared to controls, possibly due to significantly decreased amounts of yeast [29]. Patients with reduced yeast counts were significantly correlated with lower ALSFRS-R scores and forced vital capacity percentage (FVC%). Thus, shifts in various microbial populations, not only bacteria, may impact disease progression or clinical manifestation. Consistent with these findings, Zhai et al. [30] showed ALS stool samples had a significant increase in the relative abundance of Euryarchaeota (phylum), Methanobacteria (class), and *Methanobrevibacter* (genus) while *Faecalibacterium* (genus) and *Bacteroides* (genus) were reduced. ALS patients were also noted to have less rich and even microbial communities compared to healthy controls [30], which could impact metabolic function. Indeed, spectrophotometry showed that elevated fecal levels of SCFAs, nitrogen-containing compounds (NO_2_-N/NO_3_-N), and γ-aminobutyric acid were observed in ALS samples compared to controls. While not statistically significant, alterations in fecal metabolites may be a sign of underlying ALS-related GI dysfunction [30].

Given that caregivers who live or closely interact with ALS patients may share environmental exposures, and changes to the gut microbiome by extension, several studies use healthy family members and spouses as controls. Niccolai et al. [31], for example, compared the stool samples of ALS patients with those collected from cohabiting controls and found they were distinct. Patient samples showed a significant enhancement in the relative abundance of *Senegalimassilia* (genus), while *Subdoligranulum* (genus) and several members of *Lachnospiraceae* (family) were instead only elevated in family members. Among others, *Adlercreutzia*, *Lachnospiraceae_FCS020_group*, and *Romboutsia* were further correlated with disease progression and survival in the ALS group [31]. Patients with a slow progression rate, in particular, were associated with a higher abundance of *Streptococcaceae* (family) but a significant reduction in fecal α-diversity [31]. In contrast, Ngo et al. [32] reported no significant differences were found in the fecal microbiota of ALS subjects compared to the control group, which included healthy spouses, friends, and family members. Moreover, no associations between the gut microbiome and disease severity or site of onset were found. Anthropometric, metabolic, and clinical features of the disease were all described as being independent of the gut’s microbial composition [32]. Consistent with the previous study, however, the authors observed that an increase in the α-diversity of fecal microbiomes was associated with accelerated disease progression and a greater risk of early death [32]. This finding suggests that the contribution of gut dysbiosis to ALS pathogenesis may extend beyond changes to microbial composition to involve alterations in community diversity as well. 

While most human microbiome studies present evidence of gut dysbiosis in ALS patients, some conflicting findings have been reported. Inconsistencies with regards to differences in the alpha/beta-diversity or Firmicutes/Bacteroidetes (F/B) ratio are amongst the most common. The F/B ratio is widely accepted as a sign of intestinal homeostasis and overall gut health, and its alteration has been previously described in inflammatory bowel diseases [33,34] as well as metabolic disorders [35,36,37]. However, the precise implications or effects of F/B imbalance on disease progression and patient outcome in ALS remain unknown due to conflicting results amongst current profiling studies. Some studies have reported that ALS patients exhibit significant shifts in the gut microbiome in favor of either Firmicutes [30,32,38] or Bacteroides [28,31,39,40], while others failed to detect any difference in the F/B ratio between ALS patients and controls [41]. These inconsistencies likely arise due to several reasons: heterogeneity in the subtypes, segment onset, and disease stage of ALS can all lead to varying differences in the gut microbial composition among study participants, while patient demographics, limited sample sizes, and methodological differences may also influence results. Addressing these limitations in future studies is pivotal to understanding the complex role of gut dysbiosis in ALS pathogenesis. 

All currently published oral and fecal microbiome studies performed on human ALS patients are summarized in Table 1. It is important to note that evidence of dysbiosis and microbial translocation has also been found in other primary samples and/or tissues [42,43,44]. For example, in a recently published study by Liu et al. [42], nasal swabs collected from 66 ALS patients and 40 healthy caregivers differed significantly in microbial composition and diversity. At the phylum level, Bacteroidetes and Firmicutes were substantially increased in healthy nasal communities, while Actinobacteria dominated the nasal microbiota of ALS patients. Specific genera like *Gaiella*, *Sphingomonas*, *Lachnospiraceae*, and *Klebsiella* were all found to be significant predictors of ALS [42]. *Faecalibacterium* and *Alistipes* were also notably enriched in ALS patients and positively correlated with ALSFRS-R scores [42]. Further profiling and correlation analyses linked several changes to the nasal microbiota with ALS-related immune dysregulation and metabolic dysfunction. However, it is still unknown whether these associations are causal and if they significantly contribute to disease development. Another study by Ellis et al. [43] sought to characterize the microbial profile and total DNA content in the peripheral blood of ALS patients. Compared to healthy controls and multiple sclerosis (MS) patients, samples drawn from ALS subjects exhibited a significant reduction in *Pseudomonas*, *Acidovorax*, and *Acinetobacter* levels, with complete depletions of *Funneliformis* and *Cloacibacterium* [43]. *Hydrurus*, a freshwater alga belonging to the phylum Ochrophyta, was unexpectedly enriched in all ALS samples but only a quarter of the control and MS sample pools [43]. Moreover, principal component analysis revealed the combination of β-proteobacteria, γ-proteobacteria, and Ochrophyta could effectively sort and distinguish ALS patients from other sample populations.

## 4. Gut Dysbiosis Contributes to Pathology in Amyotrophic Lateral Sclerosis

Changes to the gut microbiota can influence distant organs by direct (e.g., translocation) and indirect (e.g., immune dysregulation) means. While the precise mechanisms are still under investigation, several connections between gut dysbiosis and ALS pathology have been identified (see Figure 1). It is important to highlight that further research is needed to definitively determine whether these findings are causal. 

### 4.1. Dysbiosis and Intestinal Barrier Integrity

The intestinal mucosa and its components serve as the gut’s frontline defense system against invasion by harmful pathogens and toxins. Healthy microbiomes are essential to the function and integrity of this barrier [49]. A common way by which commensal microorganisms fortify the gut’s lining mucosa is by direct upregulation of intercellular junctions. Alvarez et al. [50] demonstrate that several commensal strains of *Escherichia coli* can promote the translation and redistribution of tight junction proteins, such as zonula occludens (ZO)-1 and claudin-2. Similarly, Karczewski et al. [51] show that *Lactobacillus plantarum* can upregulate the expression of scaffold and transmembrane proteins in healthy subjects, even reversing their chemically induced dislocation in an in vitro model of human gut epithelium.

Alterations in the microbial composition of the gut can disrupt intercellular junctions, and by extension, increase mucosal permeability (a phenomenon commonly referred to as “leaky gut”). A recent study by Wu et al. [52] shows that gut dysbiosis was associated with compromised barrier integrity in a transgenic mouse model of ALS. Compared to wild-type mice, the relative abundance of *Fermicus*, *Escherichia coli*, and *Butyrivibrio Fibrisolvens*, a butyrate-producing bacteria, were all markedly lower in transgenic mice. These shifts occurred before ALS symptom onset (at 2 months of age) and were associated with a significant reduction in colonic expression of tight (ZO-1) and adherent (E-cadherin) junction proteins [52]. Moreover, levels of interleukin (IL)-17, a proinflammatory mediator of the host defense against extracellular pathogens [53], were significantly enhanced in both the intestine and blood of transgenic mice but not wild-type mice, indicating that intestinal permeability was affected. Indeed, a permeability assay revealed a two-fold increase in the serum intensity of fluorescein isothiocyanate (FITC)–dextran in ALS transgenic mice compared to wild-type, confirming that barrier integrity was altered by dysbiosis. While studies on ALS patients are lacking, evidence of compromised barrier integrity can be drawn from reports of microbial translocation and circulating inflammatory marker levels. For example, Zhang et al. demonstrate [54] that plasma levels of bacterial lipopolysaccharide (LPS), a pro-inflammatory glycolipid on the surface of most gram-negative bacteria, were significantly elevated in patients with sporadic ALS compared to healthy controls. In addition, LPS levels were negatively correlated with patient ALSFRS-R scores, suggesting that barrier integrity could influence disease progression or severity. Indeed, chronic elevation of LPS can trigger low-grade systemic inflammation, as evidenced by widespread monocyte activation [54], and subsequently impact vagal afferent signaling in the GBA [55]. Moreover, endotoxemia secondary to a leaky gut can potentially propagate to the CNS across damaged blood–brain barriers [56] and further aggravate neuroinflammation by promoting microglial overactivation or the production of reactive oxygen species [57]. Consistent with these findings, the restoration of intestinal barrier integrity by reversing dysbiosis not only suppressed local immune receptor signaling, but also alleviated neuroinflammation [58]. Another study by Kim et al. [38] showed that lipopolysaccharide-binding protein (LBP), a surrogate marker of microbial translocation [59], was significantly elevated in the plasma of ALS patients and similarly correlated with symptom severity. Moreover, an increased abundance of microbial species in the blood of these patients was seen by 16S rDNA quantification, highlighting the link between dysbiosis and increased gut permeability [38].

### 4.2. Dysbiosis and Metabolic Dysfunction

Motor neurons are remarkably vulnerable to systemic and cellular disturbances in energy homeostasis. Impaired mitochondrial function, oxidative stress, and altered glucose metabolism have therefore all been implicated in the pathogenesis of ALS [60]. Given that the gut microbiota significantly regulates nutrient availability and bioenergetics [61], recent evidence suggests that dysbiosis may drive metabolic dysfunction in ALS. Sagi et al. [62], for example, report that, in mice lacking the antioxidant enzyme superoxide dismutase 1 (SOD1), a well-established animal model of ALS, changes to the gut microbiota and F/B ratio were associated with significant metabolic dysfunction. Increased oxidative stress caused by SOD1 deficiency not only suppressed hepatic gluconeogenesis but also promoted lipid accumulation (causing fatty liver in young 15-week-old mice). Moreover, redox imbalance was associated with the increased nitrosylation and subsequent inactivation of glyceraldehyde-3-phosphate dehydrogenase (GAPDH), a crucial enzyme in glycolysis [62]. If not compensated, chronic shifts in carbohydrate metabolism can have detrimental impacts on energy homeostasis and disease progression in ALS [63]. Another study by Blacher et al. [22] demonstrates that, in both human ALS patients and SOD1 transgenic mice with glycine substituted to alanine at position 93 (G93A), significant reductions in the relative abundance of *Akkermansia muciniphila* were associated with lower levels of nicotinamide. Nicotinamide is the precursor to coenzymes that are crucial in cellular signaling, energy metabolism, and redox homeostasis [64], and its depletion was associated with aggravated motor dysfunction in mice and lower ALSFRS scores in patients [22]. These results were further confirmed in G93A mice when systemic supplementation with nicotinamide significantly ameliorated motor symptoms and prolonged survival, likely due to the restoration of mitochondrial and antioxidant functions (as revealed by a gene-ontology enrichment analysis). Another relevant metabolite is butyrate, a natural byproduct of dietary fiber fermentation in the colon. In addition to being the gut’s primary source of energy, butyrate has several immunomodulatory and metabolic functions throughout the body [65,66,67]. Moreover, butyrate may exert neuroprotective effects, as its supplementation in a motor neuron-like cell model of ALS significantly restored mitochondrial respiratory capacity and biogenesis [68]. Many of the previously discussed human [28,31,47] and animal [52,69] profiling studies report that the levels of butyrate-producing bacteria are significantly decreased in ALS. Hertzberg et al. [46] also show that ALS patients typically lack enzymes needed for butyrate metabolism, even without deficiency in these microbes.

### 4.3. Dysbiosis and Immune Dysregulation

Dysregulation of both central and peripheral immune systems has been previously described in ALS [70]. Persistent inflammation in the brains and spinal cords of ALS patients is typically associated with an increase in the number of reactive microglia and astrocytes. While initially neuroprotective [71], chronic glial cell activation exacerbates inflammation by promoting inflammasome formation and the production of several proinflammatory cytokines, such as IL-1β and IL-18 [70,72]. Impaired autophagy and the loss of metabolic support normally provided by astrocytes can also contribute to neuronal cell injury and degeneration in ALS [72]. Although many of the previously discussed profiling studies link alterations in the gut microbiota to neuroinflammation in ALS, the precise mechanisms are not fully understood. Some studies suggest immune dysregulation following dysbiosis is an indirect consequence of endotoxemia and an altered SCFA metabolism. Among its many anti-inflammatory activities, butyrate can inhibit histone deacetylase and subsequently shift microglial polarization towards an anti-inflammatory and more neuroprotective phenotype [73,74]. It can also suppress IL-17 to maintain the balance between regulatory T cells (Treg) and T helper 17 cells (Th17) for systemic immune response control [75]. In ALS patients, the levels of butyrate-producing bacteria are reported to be significantly lower compared to healthy controls [47]. These alterations not only impact SCFA production but can also exacerbate local gut inflammation and trigger a systemic or neuroinflammatory response. Indeed, Niccolai et al. [31] reported that inflammatory biomarkers such as macrophage inflammatory protein-1 alpha (MIP-1α), monocyte chemoattractant protein-1 (MCP-1), IL-1α, IL-6, IL-18, and IL-27 were significantly higher in ALS stool samples. Moreover, elevated levels of circulating inflammatory cytokines, such as IL-17 and IL-23, have been described in the serum and cerebrospinal fluid of ALS patients [76], indicating possible Treg/Th17 imbalance. Other mechanisms by which dysbiosis may contribute to immune dysregulation in ALS include disabled autophagy and increased immune cell infiltration [77], although further investigation is warranted.

## 5. Gut-Microbiota-Modulating Agents in Amyotrophic Lateral Sclerosis

Given that GBA dysfunction and dysbiosis play a significant role in the pathogenesis of ALS, interventions and treatment strategies that target the gut microbiota may be of therapeutic benefit. Gut-modulating agents are broadly categorized into four main types: prebiotics, probiotics, postbiotics, and synbiotics. The use of these compounds in neurodegenerative disorders, such as Alzheimer’s disease (AD) and Parkinson’s disease (PD), have previously proven effective in reversing dysbiosis and alleviating disease symptoms [78,79,80,81]. The following sections summarize all currently available preclinical evidence on the use of gut-microbiota-modulating agents in ALS.

### 5.1. Prebiotics

Prebiotics include a wide range of dietary fibers and compounds that can effectively stimulate the growth or activity of beneficial gut microbes [82,83]. Although initially indigestible, prebiotics undergo fermentation in the colon to produce metabolites (e.g., SCFAs) that help regulate systemic metabolism and promote overall gut health [84]. In a recent study by Zhang et al. [85], oral administration of the prebiotic galacto-oligosaccharides (GOS) not only increased the relative abundance of *Lactobacillus* but also attenuated neuroinflammation and cognitive impairment in transgenic AD mice. The use of fructo-oligosaccharides (FOS), alone or in combination with GOS, similarly promoted *Bifidobacterium* growth and alleviated AD pathology. The effects of these prebiotics were partly attributed to the downregulation of signaling pathways shared between the colons and cortices of mice [85], suggesting that GBA modulation could significantly influence CNS pathology. Preclinical evidence on the use of prebiotics in ALS remains preliminary and inconclusive. Song et al. [86], for example, demonstrated that treatment of SOD1-G93A transgenic mice with GOS-rich yogurt can significantly alter ALS progression and prolong animal lifespan. GOS administration not only rescued mitochondrial activity in skeletal muscles, it but also attenuated their denervation and atrophy. Moreover, the observed reduction in motor neuron degeneration was attributed to a marked suppression in neuroinflammation following treatment [86]. Compared to mice groups fed with normal saline or milk, yogurt consumption was associated with significantly reduced microglial and astrocyte activation, as well as lower levels of inflammatory and apoptosis-related factors. Another study by Yip et al. [87] showed that the use of eicosapentaenoic acid (EPA), an omega-3 polyunsaturated fatty acid, did not offer any therapeutic benefit in ALS. Although EPA treatment significantly reversed microglial cell activation similar to GOS, it shortened the lifespan in mice and failed to alter motor neuron loss [87]. Furthermore, an increase in neurotoxic byproducts, such as microglial 4-hydroxy-2-hexenal, was found in the spinal cords of mice treated with EPA. Further research is needed to conclusively determine whether prebiotic compounds are safe and effective.

### 5.2. Probiotics

Probiotics comprise a number of live microorganisms, often beneficial gut bacteria and yeast, that play a crucial role in maintaining the body’s microbial balance and promoting well-being [88]. Many of the currently well-recognized probiotics belong to the *Lactobacillus*, *Bifidobacterium*, *Saccharomyces*, *Enterococcus*, and *Streptococcus* genera [88]. Several studies have explored the potential of specific strains in treating neurodegenerative disorders. For example, Zhu et al. [89] showed that the administration of *Bifidobacterium breve* significantly attenuated neuroinflammation, amyloid deposition, and cognitive impairment in APP/PS1 transgenic mice. These changes were associated with increased regulation of gut microbiota composition and improvement in intestinal barrier function [89], highlighting the therapeutic potential of GBA modulation. Another study similarly reported *Bifidobacterium breve* supplementation in amyloid beta precursor protein (APP) knock-in mice increased the bioavailability of anti-oxidative metabolites, which improved cognitive function [90]. Huang et al. [91] demonstrated that the use of *Lactobacillus plantarum* can delay AD progression by regulating gliosis and tau hyperphosphorylation following a reduction in propionic acid levels. Yang et al. [92] show treatment with ProBiotic-4, a combination of *Bifidobacterium lactis*, *Lactobacillus casei*, *Bifidobacterium bifidum*, and *Lactobacillus acidophilus*, improved cognitive function and memory deficits in aged SAMP8 mice. Moreover, the combination significantly attenuated age-related disruption of the blood–brain barrier, and it also reduced cerebral neuronal and synaptic injuries [92]. Sancandi et al. [93] reported that Symprove™, a commercially available probiotic suspension comprised of another four bacterial strains, restored gut integrity, improved SCFA production, and prevented striatal neuroinflammation in an early stage PD rat model. 

Aligned with the aforementioned findings, probiotic interventions in ALS have been shown to be neuroprotective. A recent study by Labarre et al. [94] investigated the effects of sixteen different probiotic formulations on *Caenorhabditis elegans* strains that were genetically modified to express two human ALS-associated proteins: fused in sarcoma (FUS) and TAR DNA-binding protein 43 (TDP-43). Although most combinations had little to no effect, treatment with the probiotic *Lacticaseibacillus rhamnosus* HA-114 alone was effective in delaying neurodegeneration and preventing paralysis [94]. These effects were observed in FUS and TDP-43 mutant worms but not wild-type strains, suggesting that the benefits of HA-114 were more specifically tied to ALS pathology. Indeed, further investigations revealed that impaired β-oxidation and altered energy homeostasis, common features of ALS-related metabolic dysfunction, exacerbated motor neuron degeneration unless treated with HA-114 [94]. The neuroprotective mechanisms of *Lacticaseibacillus rhamnosus* were primarily attributed to its unique fatty acid content, which helped restore energy metabolism independent of a functional carnitine shuttle. Blacher et al. [22] similarly reported that restoration of mitochondrial function following administration of *Akkermansia muciniphila* was central to alleviating motor symptoms in SOD1-G93A mice. Although further investigation is warranted, these findings suggest that targeting dysbiosis to address neuroinflammation and metabolic imbalances in ALS could be a viable therapeutic approach.

### 5.3. Postbiotics

Postbiotics are non-viable, biologically active components or metabolic byproducts of the gut microbiota [84]. Common examples of postbiotics are functional proteins, extracellular polysaccharides, bacterial lysates, and fermentation byproducts (e.g., SCFAs). In ALS, Zhang et al. [69] have shown that the natural bacterial product, butyrate, was neuroprotective when given at a 2% concentration in filtered drinking water to SOD1-G93A transgenic mice. Treatment with postbiotics not only restored intestinal microbial homeostasis and gut barrier integrity, but also delayed ALS progression and prolonged the lifespan of mice. Moreover, abnormal Paneth cell accumulation and SOD1 mutant protein aggregation were significantly lowered in the intestines of mice receiving butyrate compared to the control groups [69]. These findings suggest that the therapeutic potential of postbiotics particularly lies in their ability to address gut-related abnormalities and inflammation. Indeed, Ogbu et al. [95] demonstrated that 2% sodium butyrate administration was associated with changes in microbial carbohydrate and amino acid metabolism. Moreover, butyrate treatment significantly reduced microglia in the spinal cords of SOD1-G93A mice and was associated with lower circulating levels of proinflammatory IL-7 and LPS [95]. Ryu et al. [96] further attributed the neuroprotective effects of sodium phenylbutyrate to the regulation of several anti-apoptotic genes. In addition to inhibiting histone deacetylase, phenylbutyrate administration significantly upregulated nuclear factor-kappaB (NF-kB) and beta-cell lymphoma 2 (Bcl-2) expression, blocking caspase activation and subsequent motor neuron death. While treatment with phenylbutyrate alone significantly delayed disease progression, another study by Signore et al. [97] reported that its combination with riluzole, a widely recognized medication for the treatment of ALS patients, was most effective in prolonging survival in G93A transgenic ALS mice. The combination also rescued body weight loss and grip strength [97], features of the disease that are often overlooked in animal studies. While promising, larger-scale studies are required to determine the long-term efficacy of combined therapies and whether these synergistic effects translate to human patients. All of the aforementioned findings on the application of gut-modulating agents in ALS have been summarized in Table 2.

## 6. Future Directions

Taken all together, recent evidence suggests a role for gastrointestinal dysfunction and gut dysbiosis in the pathogenesis of neurodegenerative disorders. Human microbiome studies consistently report that ALS patients exhibit distinct changes to their gut microbial composition and diversity. Drastic shifts in the microbial profile not only exacerbate local intestinal inflammation but can also promote chronic neuroinflammation, a hallmark of ALS pathology. Moreover, compromised gut barrier integrity, metabolic dysfunction, and immune dysregulation following dysbiosis have been linked to GBA dysfunction in ALS. Given these connections, the therapeutic potential of gut-modulating agents (prebiotics, probiotics, and postbiotics) have become a focus of recent research efforts. These interventions showed promising gut modulatory effects and neuroprotective properties in animal models of ALS. Depending on when they were administered, several agents were also able to rescue motor symptoms, or even mitigate disease progression. Although many parameters of gut health were assessed (e.g., fecal metabolites, inflammatory markers, and intestinal permeability), a major limitation of the preclinical studies summarized in this review is that the composition of gut microbiota following treatment was not always reported. It is, therefore, difficult to determine whether the effects of gut-modulating agents are secondary to the restoration of a healthy microbial balance without further research. Future studies should also continue exploring ALS-related gut dysbiosis to establish which mechanisms are potentially causal to disease pathology. To limit the possibility of any confounding factors from independently affecting both ALS and the gut microbiome, rigorous study designs are necessary. Lastly, translating these findings to human patients remains a challenge. Preclinical studies often use transgenic mice with SOD1 mutations, and while instrumental in our understanding of ALS pathology, these models do not fully represent the ALS patient population. It is important to investigate whether neuroprotection following gut modulation is reproducible in a variety of genetic and sporadic ALS models. Well-designed, large-scale randomized clinical trials are needed to better determine the efficacy and safety of these agents in ALS patients. Future studies should also aim to explore how individual variations in the human gut microbiome impact disease progression and treatment outcomes. Understanding the influence of genetic, dietary, and environmental factors could pave the way for more personalized treatment strategies.

## Figures and Tables

**Figure 1 nutrients-16-00590-f001:**
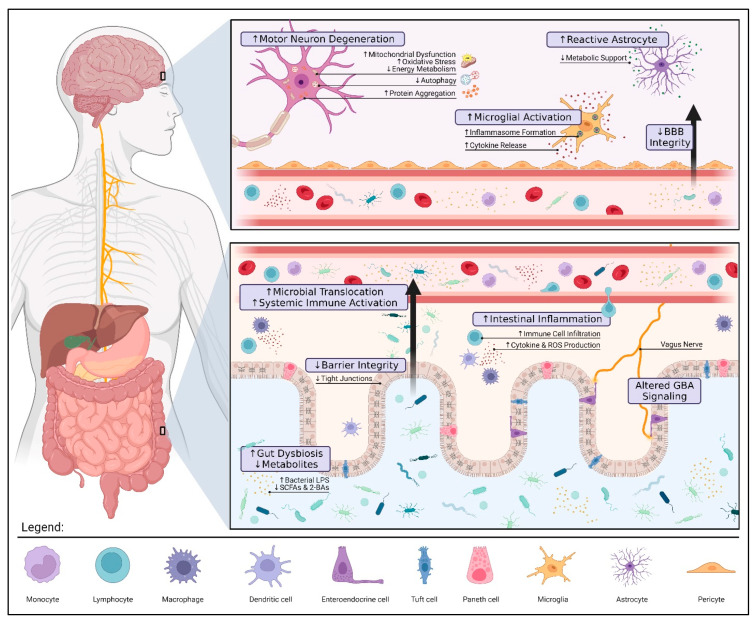
**Gut Dysbiosis and Pathology in Amyotrophic Lateral Sclerosis.** Alterations to the gut microbiota can contribute to ALS pathology via three pivotal mechanisms: compromised gut barrier integrity, metabolic dysfunction, and immune dysregulation. Reduced expression of tight junctions along the intestinal epithelium allows for microbial invasion and subsequent inflammation. These microbes may also translocate into the blood, triggering a systemic immune response (e.g., endotoxemia, proinflammatory cytokine production, and peripheral monocyte activation). If prolonged, systemic inflammation can damage the blood–brain barrier (BBB) and result in the overactivation of microglia and astrocytes further aggravating neuroinflammation. The loss of immunomodulatory and neuroprotective metabolites such as butyrate, a short-chain fatty acid (SCFA), promotes motor neuron degeneration by increased oxidative stress and mitochondrial dysfunction. The interplay between gut and brain health highlights the therapeutic potential of gut–brain axis (GBA) modulation in ALS. Restoring a healthy microbial balance may not only alleviate patient symptoms by modulating these mechanisms but also prolong survival by mitigating disease progression. The figure was created using Biorender.com.

**Table 1 nutrients-16-00590-t001:** Key Microbial Alterations in Human Patients with ALS.

Author	Sample Size	Method of Assessment	Significant Microbial Changes	Outcomes
Gong et al., 2023 [45]	70 subjects: 35 ALS and 35 age/sex-matched healthy controls	Stool samples analyzed via 16S rRNA sequencing Fecal metabolites evaluated via mass spectrometry	↑ *Bacteroidia* and *Verrucomicrobiae* (in ALS patients with cognitive impairment vs. without) ↓ *Proteobacteria*	Microbial β-diversity and the F/B ratio were significantly different in ALS patients with vs. without cognitive impairment Altered bile acid metabolism was associated with greater cognitive decline
Guo et al., 2023 [26]	185 subjects: 75 ALS and 110 healthy controls (unrelated)	Stool samples analyzed via 16s rRNA sequencing Plasma metabolites evaluated via mass spectroscopy Plasma metabolite-gut microbiome associations evaluated via WGCNA and O2PLS-DA	↑ *Bacteroides*, *Parasutterella*, and *Lactococcus* ↓ *Faecalibacterium* and *Bifidobacterium*	Altered lipid metabolism (mostly acylcarnitines) was significantly correlated with ALSFRS-R scores and disease progression
Kim et al., 2022 [38]	56 subjects: 36 ALS and 20 healthy controls (spouses)	Stool and saliva samples analyzed via qPCR and 16S rRNA sequencing Blood microbial translocation evaluated via qPCR and plasma LBP levels	↑ *Ruminococcaceae* (stool), *Prevotellaceae* (saliva), and *Fusobacteria* (stool and saliva) ↓ *Bacteroidaceae* (stool) and *Veillonellaceae* (saliva)	sALS patients demonstrated significant gut dysbiosis and a high fecal F/B ratio bALS patients demonstrated significant oral dysbiosis and a low oral F/B ratio High levels of dysbiosis were correlated with greater microbial translocation and symptom severity
Niccolai et al., 2021 [31]	28 subjects: 19 ALS and 9 sex-matched healthy controls (family caregivers)	Stool samples analyzed via 16S rRNA sequencing Fecal and serum cytokine expression evaluated via multiplex immunoassay Fecal metabolites evaluated via mass spectrometry	↑ *Senegalimassilia* ↓ *Monoglobaceae*, *Erysipelatoclostridiaceae*, *Clostridiaceae*, *Adlercreutzia*, *Monoglobus*, *Lachnospiraceae*, *Fusicatenibacter*, *Marvinbryantia*, and *Subdoligranulum*	ALS patients had a lower F/B ratio compared to controls ALS patients had lower amounts of IL-8, IL-15, VEGF-A, and MCP-1 compared to controls
Hertzberg et al., 2021 [46]	40 subjects: 10 ALS and 30 healthy controls (10 spouses and 20 unrelated)	Rectal swabs analyzed via 16S rRNA sequencing Fecal and plasma inflammatory markers evaluated via ELISA and other immunoassays	↓ *Prevotella timonensis*	ALS patients had more rich and even microbial communities compared to their spouses No difference in inflammatory marker expression between ALS patients and their spouses was found ALS patients lacked key enzymes in butanoate metabolism
Nicholson et al., 2021 [47]	139 subjects: 66 ALS, 12 neurodegenerative controls, and 61 healthy controls	Stool samples analyzed via 16S rRNA sequencing and shotgun metagenomic sequencing	↑ *Prevotella copri*, *Phascolarctobacterium succinatutens*, *Bacteroides clarus*, *Dorea* (unclassified), and *Escherichia* (unclassified) ↓ *Adlercreutzia equolifaciens*, *Lachnospiraceae bacterium*, *Peptostreptococcaceae* (unclassified), *Coprobacter fastidiosus*, *Ruminococcus lactaris*, *Eubacterium eligens*, *Bifidobacterium longum*, *Roseburia intestinalis*, and *Eubacterium rectale*	Higher abundance of butyrate-producing bacteria lowers the risk of developing ALS
Di Gioia et al., 2020 [29]	100 subjects: 50 ALS and 50 age/sex-matched healthy controls (unrelated)	Stool samples analyzed via qPCR, PCR-DGGE, and 16S rRNA sequencing	↑ *Escherichia coli*, *Clostridiales Family XI*, *Gastranaerophilalaes*, and Cyanobacteria ↓ *Clostridiaceae 1* and yeasts	No difference in total bacteria between ALS patients and controls was found Microbial composition in ALS patients was noted to change with time (particularly Bacteroidetes)
Ngo et al., 2020 [32]	100 subjects: 49 ALS and 51 age/sex/BMI-matched healthy controls (spouses, friends, or family members)	Stool samples analyzed via 16S rRNA sequencing, direct taxa comparison, and predictive metagenomics	No significant microbial changes	ALS patients with a highly diverse fecal microbiome or a high F/B ratio were at greater risk of disease acceleration and early death
Zeng et al., 2020 [39]	40 subjects: 20 ALS and 20 age-matched healthy controls (unrelated)	Stool samples analyzed via 16S rRNA sequencing, metabolomic analysis, and shotgun metagenomic sequencing	↑ Bacteroidetes, *Kineothrix*, *Parabacteroides*, *Odoribacter*, *Sporobacter*, *Eisenbergiella*, *Mannheimia*, *Anaerotruncus*, *Porphyromonadaceae* (unclassified), *Sulfuricurvum kujiense*, *Cyanothece* sp., and *Haladaptatus paucihalophilus* ↓ Firmicutes, *Megamonas*, and *Enterococcus columbae*	ALS patients had a lower F/B ratio compared to controls Microbial alterations in ALS patients were linked with disruption or decline of several metabolic and intracellular pathways
Blacher et al., 2019 [22]	66 subjects: 37 ALS and 29 age/BMI-matched healthy controls (family members)	Stool samples analyzed via shotgun metagenomic sequencing Fecal and serum metabolites evaluated via mass spectrometry	↑ *Escherichia coli* and *Oscillibacter* (unclassified) ↓ *Anaerostipes hadrus*, *Bacteroidales bacterium ph8*, *Bifidobacterium pseudocatenulatum*, *Eubacterium hallii*, *Lachnospiraceae bacterium*, *Eubacterium rectale*, *Eubacterium ventriosum*, *Roseburia hominis*, and *Faecalibacterium prausnitzii*	ALS patients exhibited decreased expression of genes involved in metabolism of tryptophan and NAM Higher serum levels of NAM were significantly correlated with better scores on the ALSFRS-R
Zhai et al., 2019 [30]	16 subjects: 8 ALS and 8 healthy controls	Stool samples analyzed via 16S rRNA sequencing Fecal metabolites evaluated via ELISA and spectrophotometry	↑ Euryarchaeota, Methanobacteria, and *Methanobrevibacter* ↓ *Faecalibacterium* and *Bacteroides*	ALS patients had less rich bacterial communities compared to controls which may impact metabolic function ALS patients had a higher F/B ratio compared to controls
Brenner et al., 2018 [41]	55 subjects: 25 ALS and 32 age/sex-matched healthy controls	Stool samples analyzed via qRT-PCR, 16S rDNA sequencing, and predictive metagenomics	No significant microbial changes apart from differences in the proportions of uncultured *Ruminococcaceae*	No difference in F/B ratio or predicted metagenome between ALS patients and controls was found
Mazzini et al., 2018 [48]	100 subjects: 50 ALS and 50 age/sex-matched healthy controls	Stool samples analyzed via qPCR and PCR-denaturing gradient gel electrophoresis	Significance was not commented on, but relative abundance: ↑ *Escherichia coli* and *Enterobacteria* ↓ *Clostridium* and yeast	ALS patients had a lower DNA concentration compared to healthy controls
Rowin et al., 2017 [40]	101 subjects: 4 ALS, 1 BAD, and 96 healthy controls	Stool samples analyzed via 16S rDNA sequencing, bacterial and mycological culture, parasitology tests, and enzyme immunoassay Fecal metabolites evaluated via mass spectrometry	Significance was not commented on, but relative abundance: ↑ *Bacteroides-Prevotella group*, *Odoribacter* spp., *Barnesiella* spp., and *Bacteroides vulgatus* ↓ *Ruminococcus* spp., *Clostridium* spp., and *Roseburia* spp.	Three of four ALS patients showed elevation in stool inflammatory biomarkers ALS patients with low levels of *Ruminococcus* spp. had lower F/B ratios
Fang et al., 2016 [28]	11 subjects: 6 ALS and 5 healthy controls	Stool samples analyzed via 16S rRNA sequencing	↑ Bacteroidetes, Bacteroidia, Bacteroidales, and *Dorea* ↓ Firmicutes, Clostridia, *Lachnospiraceae*, *Oscillibacter*, and *Anaerostipes*	ALS patients had a lower F/B ratio compared to controls

Note: Only oral and fecal microbiome studies of human patients were considered and summarized in this table. Upward arrows (↑) indicate an increase in the abundance of listed microbes while downward arrows (↓) indicate reduced levels. Unless mentioned otherwise, all listed microbial changes in ALS patients compared to controls are statistically significant (*p* < 0.05). ALSFRS-R: ALS functional rating scale revised; BAD: brachial amyotrophic diplegia; bALS: bulbar onset ALS; sALS: spinal onset ALS; DGGE: denaturing gradient gel electrophoresis; F/B: Firmicutes/Bacteroidetes ratio; IL: interleukin; LBP: lipopolysaccharide-binding protein; MCP-1: monocyte chemoattractant protein-1; NAM: nicotinamide; PCR: polymerase chain reaction; qPCR: quantitative PCR; qRT-PCR: quantitative real-time reverse-transcription PCR; VEGF-A: vascular endothelial growth factor A, O2PLS-DA: two-way orthogonal partial least square with discriminant analysis; WGCNA: weighted gene co-expression network analysis.

**Table 2 nutrients-16-00590-t002:** Preclinical Studies on Gut-Modulating Agent Use in Animal Models of ALS.

Author	Intervention	Type	Animal Model	Outcome
Song et al., 2013 [86]	Treatment with galactooligosaccharide and galactooligosaccharide-rich prebiotic yogurt	Prebiotic	SOD1-G93A transgenic mice	Elevated vitamin B9 (folate) and B12 (cobalamin) levels but reduced homocysteine levels Suppressed neuroinflammation and apoptosis which drastically reduced motor neuron loss Delayed disease progression, prolonged lifespan, and improved symptoms in ALS mice
Yip et al., 2013 [87]	Treatment with Incromega SE7010R oil enriched in eicosapentaenoic acid, an omega-3 polyunsaturated fatty acid	Prebiotic	SOD1-G93A transgenic mice	Reduced microglial and astrocyte activation, but increased toxic byproducts of omega-3 fatty acids Accelerated disease progression and reduced lifespan in ALS mice
Labarre et al., 2022 [94]	Treatment with thirteen individual bacterial strains (including *Bacillus subtilis*, *Bifidobacterium breve*, *Bifidobacterium animalis* subsp. *lactis*, *Lacticaseibacillus plantarum*, *Lacticaseibacillus casei*, *Lacticaseibacillus paracasei*, *Lacticaseibacillus helveticus*, *Lacticaseibacillus rhamnosus*, and *Pediococcus acidilactici*) or three combinations	Probiotic	*Caenorhabditis elegans*	*L. rhamnosus* HA-114 regulated genes involved in mitochondrial β-oxidation and branch-chain amino acid breakdown *L. rhamnosus* HA-114 was the most effective at attenuating neurodegeneration and rescuing motor phenotype
Blacher et al., 2019 [22]	Treatment with one bacterial strain (*Akkermansia muciniphila*)	Probiotic	SOD1-G93A transgenic mice	Increased nicotinamide levels which restored mitochondrial and antioxidant functions Alleviated motor symptoms and prolonged survival in ALS mice
Zhang et al., 2017 [69]	Treatment with sodium butyrate added at 2% concentration to filtered drinking water	Postbiotic	SOD1-G93A transgenic mice	Reduced intestinal permeability and restored levels of butyrate-producing bacteria Delayed disease onset and prolonged lifespan in ALS mice
Ogbu et al., 2022 [95]	Treatment with sodium butyrate added at 2% concentration to filtered drinking water	Postbiotic	SOD1-G93A transgenic mice	Restored a healthy metabolic state by altering carbohydrate and amino acid metabolism as well as the formation of gamma-glutamyl amino acids Reduced microglial activation and inflammatory marker levels in ALS mice
Ryu et al. 2005 [96]	Treatment with phenylbutyrate dissolved in phosphate-buffered saline	Postbiotic	G93A transgenic mice	Inhibited histone deacetylase (shifting microglial to an anti-inflammatory neuroprotective phenotype) and upregulated expression of anti-apoptotic genes Promoted motor neuron survival and delayed disease progression in ALS mice

Note: To the best of our knowledge, no preclinical studies explored the use of synbiotics in ALS. SAMP8: senescence-accelerated mouse prone 8; SOD1: superoxide dismutase 1; G93A: glycine 93 to alanine mutation.

## Data Availability

The original contributions presented in the study are included in the article, further inquiries can be directed to the corresponding author.

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
