# Peer review of "Gut-Modulating Agents and Amyotrophic Lateral Sclerosis: Current Evidence and Future Perspectives"

_nutrients, 2024, doi:10.3390/nu16050590_

Round 1

Reviewer 1 Report

Comments and Suggestions for Authors

I congratulate the authors for their paper

my work has been facilitated by the fluent writing

I report some critical points that I think the Authors can correct to improve the quality of the paper

line 88 it would be convenient to specify that viruses are also part of it, although the subject is not dealt with

line 94 correct Cyanobacteri with Cyanobacteria

line 94-99 I think you can provide a more recent reference for this statement. Furthermore, you can add a sentence about quick changes of microbiota following diet or other modifications in lifestyle, etc.

line 125 the phrase "Indeed, several microbes such as Akkermansia muciniphila and members of the Lachnospiraceae family were correlated with alterations in plasma metabolite levels, particularly lipids" is not clear. Do you mean lipid-related metabolites? please clarify

line 184 the phrase "some report..." can be clearer, do you mean "some researchers"?

line 266-269 Which is the reference?

line 329 missing reference

line 330 very nice image 

Comments on the Quality of English Language

Language is fluent and understandable, only some minor corrections can be provided to clarify certain steps

Author Response

We'd like to thank the reviewer for evaluating our manuscript and providing suggestions on how it can be improved. Changes to the manuscript for this reviewer are highlighted in blue.  Lines cited in the following amendments are according to the new manuscript.

#1: Viruses have been added with the appropriate citation at L88-89: “Bacteriophages and eukaryotic viruses are also integral components of the gut microbiome [15].”

#2: Cyanobacteria has been amended at L95.

#3: Two recent references describing genetic influences on microbial composition have been added at L96-99. Section 3 is more focused on presenting evidence of gut dysbiosis in ALS, but a new sentence has been added at L100-102: “Furthermore, studies have shown that dietary changes or acute lifestyle modifications can quickly and reproducibly alter the gut microbiota [20], [21], [22].”

#4: Yes. The phrase refers to lipid-related metabolites and “acylcarnitine, bile acid, and fatty acid…” are specified in the following sentence. L129-131 has been now amended to “members of the Lachnospiraceae family were correlated with alterations in plasma lipid-related metabolites [24]. Mendelian randomization analysis described acylcarnitine, bile acid, and fatty acid metabolism...”.

#5: Yes. The words ‘researchers’ and ‘studies’ are interchangeable in this case, but ‘studies’ is more consistent with the previous sentence. L187-189 has been now corrected to “…conflicting results amongst current profiling studies. Some studies report that ALS patients exhibit significant shifts in the gut microbiome in favor of either Firmicutes [28], [30], [36] or Bacteroides [26], [29], [37], [38], while others failed to…”.

#6: It refers to the study by ‘Kim et al.’ discussed in the previous sentence, but a citation would make it clearer and has been now added at L276-278.

#7: The appropriate reference has been added at L339-341.

#8: Thank you!

Reviewer 2 Report

Comments and Suggestions for Authors

This review offers a summary of studies describing the association of gut dysbiosis with ALS. The title mentions "future perspectives." I don't see any future perspectives or new direstions other than the same effort to modulate gut dysbiosis with more symptomatic treatments using probiotic/prebiotic/and other supplements (referred to as "agents). Where are the pathophysiological insights explaining potential causative mechanisms linking ALS and dysbiosis? Where is the proof of neuroprotection through gut modulations? See my comments in the attached pdf file.

Author Response

We would like to kindly thank the reviewer for their assessment of our manuscript and comments. Lines cited in the following amendments are according to the new manuscript. Changes to the manuscript for this reviewer are highlighted in yellow.  Comments on the website are addressed by points #1-3 while comments on the pdf are addressed by points #A-D.

#1: The reviewer believes no causative pathophysiological mechanisms have been presented. While it is true that the links drawn between gut dysbiosis and ALS pathology are largely putative, we have tried to maintain this throughout the manuscript. In our text, dysbiosis is described as a potential ‘contributing’ not ‘causal’ factor and this is consistent with previously published literature (see references 1-2). However, the novelty of our work compared with other reviews is in summarizing the findings of both experimental and profiling studies to delineate which specific microbial alterations contribute to the proposed mechanisms. We do acknowledge at the end of section 3 (L209) that “…it is still unknown whether these associations are causal…” and in section 4 (L229-230) that “…the precise mechanisms are still under investigation…”. The reviewer is still correct in that this point could be more explicit. Section 4 has, therefore, been amended to include the following sentence at L231-232: “It is important to highlight that further research is needed to definitively determine whether these findings are causal.”      

#2: The reviewer believes no proof of neuroprotection has been presented. Most interventions discussed in section 5 and table 2 not only alleviate disease symptoms but also modulate some aspects of ALS pathology. Prebiotics at L383-384 “…significantly reduced microglial and astrocyte activation as well as lower levels of inflammatory- and apoptosis-related factors”. Probiotics at L370-371 “…attenuated neuroinflammation …” and at L428-429 “…restore energy metabolism”. Postbiotics at L450-451 “significantly reduced microglia in the spinal cords…[and] lower circulating levels of proinflammatory IL-7 and LPS”. The authors of the cited studies in section 5 and table 2 describe the aforementioned effects as ‘neuroprotective’ and responsible for improvements in lifespan/motor function. However, a major limitation of these preclinical studies is that, while several parameters of gut health were assessed (e.g., fecal metabolites, inflammatory markers, and intestinal permeability), the composition of gut microbiota following treatment was not always reported. Only the study by Zhang et al. in Table 2 confirmed that the intervention “restored levels of butyrate-producing bacteria in the gut”. We understand why it is difficult to directly prove neuroprotection through gut modulation. These point have now been added to section 6 at L479-488.

#3: The reviewer believes the future directions section could be improved because “there are no insights into causative mechanisms for future research, only described associations and symptomatic treatments”. We hope the answers and manuscript amendments above are satisfactory to the reviewer and welcome any further suggestions.

#A: This line highlights that various microbial changes, not only bacteria, may impact patient outcomes. While the reviewer’s comment is valid, the consideration of confounding factors independently affecting both ALS and dysbiosis is a more relevant addition to the future directions. Therefore, the following amendment has been added to section 6 at L486-488: “To limit the possibility of any confounding factors from independently affecting both ALS and the gut microbiome, rigorous study designs are necessary.”

#B: The abbreviation “nitrogen-containing compounds (NO2-N/NO3-N)” has been added at L155.

#C: The following sentence at L231-232 has been added for emphasis: “It is important to highlight that further research is needed to definitively determine whether these findings are causal.”

#D: The phrase “…Alzheimer's disease (AD) and Parkinson’s disease (PD) has previously proven effective in reversing dysbiosis and alleviating disease symptoms [72], [73], [74], [75].” is not related to the reviewer’s comment. The references cited in this line also do not discuss the same agents mentioned throughout section 5 since they are for different disorders (AD/PD vs ALS). Point #2 addresses that the agents discussed in section 5 are not only symptomatic but also modulate ALS pathology, and the appropriate amendment has been added to section 6.

#E: Point #2 addresses that the little causative evidence of “neuroprotection following gut modulation” is indeed a limitation of the reported literature. Future research could benefit from this, and the appropriate amendment has been added to section 6.

References:

1) Boddy, S.L., Giovannelli, I., Sassani, M. et al. The gut microbiome: a key player in the complexity of amyotrophic lateral sclerosis (ALS). BMC Med 19, 13 (2021). https://doi.org/10.1186/s12916-020-01885-3

2) Sun J, Huang T, Debelius JW, Fang F. Gut microbiome and amyotrophic lateral sclerosis: A systematic review of current evidence. J Intern Med. 2021 Oct;290(4):758-788. https://doi.org/10.1111/joim.13336

Reviewer 3 Report

Comments and Suggestions for Authors

The manuscript is a review of the literature about the gut-modulating agents in ALS. The authors' goals are the revision of the current data and the draft of the future goals. The paper is clear and well written. I do not have specific concerns about the papers included. The figure included is clear and very helpful for the paper. I have only a few comments for the authors aimed at the improvement of the manuscript:

- have you evaluated to include in the tables a qualitative evaluation of the studies included?

- have you evaluated to include a list of abbreviations?

- I think section #2 required more details about the components

- please clarify the possible role of the intestinal barrier integrity

Author Response

We'd like to thank the reviewer for evaluating our manuscript and providing suggestions on how it can be improved. Changes to the manuscript for this reviewer are highlighted in green. Lines cited in the following amendments are according to the new manuscript.

#1: Yes. We recognize the value of a qualitative analysis, but the aim was to collect data in a concise and objective manner. This approach ensures a more standardized way of reporting study findings and allows readers to compare results more directly. Reviews on other neurodegenerative disorders similarly omit a qualitative analysis.

#2: Yes. We initially considered including a list of abbreviations but resorted to spelling them in the manuscript directly. The tables and figures also include abbreviations of their own. To make sure no abbreviations were missed, we checked the manuscript and added the following:

-“BAD: Brachial Amyotrophic Diplegia” at L223

-“DGGE: denaturing gradient gel electrophoresis” at L223

-“qPCR: quantitative polymerase chain reaction” at L224

-“VEGF: vascular endothelial growth factor A” at L225

-“nitrogen-containing compounds (NO2-N/NO3-N)” at L155

-“glycine substituted to alanine at position 93 (G93A)” at L296

-“amyloid beta precursor protein (APP)” at L404

#3: Section 2 was indeed brief and concise in its description of the GBA components. Compared to reviews that are solely focused on studying the GBA, our goal was only to give an overview/background so that any gut-brain connections discussed in Section 4 or Figure 1 are clear to readers. If more details are still necessary, please let us know so we can expand on our description of the GBA.

#4: Intestinal barrier integrity’s plausible role and contribution to ALS pathophysiology is in microbial translocation and systemic immune system activation (which, in turn, promote neuroinflammation and neuronal death). For clarification, the section has been now amended at L268-273: “Moreover, endotoxemia secondary to a leaky gut can potentially propagate to the CNS across damaged blood-brain barriers [57] and further aggravate neuroinflammation by promoting microglial overactivation or the production of reactive oxygen species [58]. Consistent with these findings, the restoration of intestinal barrier integrity by reversing dysbiosis not only suppressed local immune receptor signaling but also alleviated neuroinflammation [59].”

Round 2

Reviewer 2 Report

Comments and Suggestions for Authors

Although I am not convinced that gut modulating agents play a role in the association of ALS with dysbiosis, gut modulating agents are the focus of this review, and the topic has been adequately covered.